# A Methodological Approach to Use Contextual Factors for Epidemiological Studies on Chronic Exposure to Air Pollution and COVID-19 in Italy

**DOI:** 10.3390/ijerph19052859

**Published:** 2022-03-01

**Authors:** Lisa Bauleo, Simone Giannini, Andrea Ranzi, Federica Nobile, Massimo Stafoggia, Carla Ancona, Ivano Iavarone

**Affiliations:** 1Department of Epidemiology, Lazio Regional Health Service, ASL Roma 1, 00147 Rome, Italy; l.bauleo@deplazio.it (L.B.); f.nobile@deplazio.it (F.N.); m.stafoggia@deplazio.it (M.S.); c.ancona@deplazio.it (C.A.); 2Environmental Health Reference Centre, Regional Agency for Environmental Prevention of Emilia-Romagna, 41124 Modena, Italy; sgiannini@arpae.it; 3Department of Environment and Health, Istituto Superiore di Sanità, 00161 Rome, Italy; ivano.iavarone@iss.it

**Keywords:** contextual factors, data synthesis techniques, principal component analysis (PCA), air pollution, COVID-19, epidemiology

## Abstract

The large availability of both air pollution and COVID-19 data, and the simplicity to make geographical correlations between them, led to a proliferation of ecological studies relating the levels of pollution in administrative areas to COVID-19 incidence, mortality or lethality rates. However, the major drawback of these studies is the ecological fallacy that can lead to spurious associations. In this frame, an increasing concern has been addressed to clarify the possible role of contextual variables such as municipalities’ characteristics (including urban, rural, semi-rural settings), those of the resident communities, the network of social relations, the mobility of people, and the responsiveness of the National Health Service (NHS), to better clarify the dynamics of the phenomenon. The objective of this paper is to identify and collect the municipalities’ and community contextual factors and to synthesize their information content to produce suitable indicators in national environmental epidemiological studies, with specific emphasis on assessing the possible role of air pollution on the incidence and severity of the COVID-19 disease. A first step was to synthesize the content of spatial information, available at the municipal level, in a smaller set of “summary indexes” that can be more easily viewed and analyzed. For the 7903 Italian municipalities (1 January 2020—ISTAT), 44 variables were identified, collected, and grouped into five information dimensions a priori defined: (i) geographic characteristics of the municipality, (ii) demographic and anthropogenic characteristics, (iii) mobility, (iv) socio-economic-health area, and (v) healthcare offer (source: ISTAT, EUROSTAT or Ministry of Health, and further ad hoc elaborations (e.g., OpenStreetMaps)). Principal component analysis (PCA) was carried out for the five identified dimensions, with the aim of reducing the large number of initial variables into a smaller number of components, limiting as much as possible the loss of information content (variability). We also included in the analysis PM_2.5_, PM_10_ and NO_2_ population weighted exposure (PWE) values obtained using a four-stage approach based on the machine learning method, “random forest”, which uses space–time predictors, satellite data, and air quality monitoring data estimated at the national level. Overall, the PCA made it possible to extract twelve components: three for the territorial characteristics dimension of the municipality (variance explained 72%), two for the demographic and anthropogenic characteristics dimension (variance explained 62%), three for the mobility dimension (variance explained 83%), two for the socio-economic-health sector (variance explained 58%) and two for the health offer dimension (variance explained 72%). All the components of the different dimensions are only marginally correlated with each other, demonstrating their potential ability to grasp different aspects of the spatial distribution of the COVID-19 pathology. This work provides a national repository of contextual variables at the municipality level collapsed into twelve informative factors suitable to be used in studies on the association between chronic exposure to air pollution and COVID-19 pathology, as well as for investigations on the role of air pollution on the health of the Italian population.

## 1. Introduction

Air pollution is a major global public health risk factor and puts an enormous health and economic burden on human societies. Based on the last available estimates, air pollution ranked 4th among major mortality risk factors globally, exceeding the impacts of obesity, high cholesterol, and malnutrition. Air pollution is estimated to have contributed to 6.67 million deaths worldwide in 2019, nearly 12% of the global total, and ambient PM_2.5_ alone is responsible for 4.14 million deaths [1,2].

An overwhelming body of evidence has accumulated over the past two decades, demonstrating that health effects of air pollution can affect nearly all organ systems [3,4]. Recent systematic reviews of epidemiological evidence linking ambient air pollution (both long- and short-term exposures) to human health are collected in a Special Issue [5], adopted as a basis to inform the formulation of the new air quality guidelines (AQG) published by WHO in 2021 [6]. The new AQGs reflect the large impact of air pollution on global health, halving the recommended limits for average annual PM_2.5_ levels from 10 micrograms per cubic meter to 5, and lowering those for PM_10_ from 20 to 15 micrograms. 

There is now broad scientific consensus that long-term exposures to air pollution contribute to increased risk of illness and death from ischemic heart disease, lung cancer, chronic obstructive pulmonary disease (COPD), lower-respiratory infections (e.g., pneumonia), stroke, type 2 diabetes, and adverse birth outcomes [1,2,7,8]. Interestingly, many chronic health conditions, such as diabetes, cardiovascular disease and COPD, have also been associated with increased vulnerability to COVID-19 [1,9,10,11]. Long-term exposure to air pollution can therefore indirectly worsen the prognosis of COVID-19 by increasing the risk of chronic diseases associated with COVID-19, but can also act directly, as it can suppress or influence early immune responses to SARS-CoV-2 infection [12] and alter the host’s immunity towards respiratory infections [13]. These associations, moreover, have been shown to be biologically plausible [14]. However, the exact contribution of long-term exposure to air pollutants in modulating the spread and severity of COVID-19 is still controversial.

The availability of both atmospheric pollution and COVID-19 data, and the easiness to make simple geographical correlations between them, has led to a proliferation of ecological studies which have related the levels of pollution in an area (county, municipality, zip code areas, region, etc.) to COVID-19 incidence, mortality, or lethality rates in that area [12,15,16,17,18]. However, the potential risk of ecological fallacy can lead to non-existent risk associations or, even worse, in the opposite direction compared to true associations at an individual level. The strengths and limitations of different approaches, as well as challenges and recommendations for studying outdoor air pollution in relation to COVID-19, have been reviewed [11,19,20,21]. Studies available on the incidence, spread and severity of COVID-19 have not taken into account, or have not done so adequately, individual risk factors such as gender, age, area of residence, comorbidities, or occupation, as well as the role of context variables, such as socio-economic deprivation, health supply, production activities that may involve a greater risk of contagion, social interactions in the community, mobility, time-activity patterns, type of environment (urban, rural, semi-rural) and demographic factors [20]. 

Due to the limitations of the data currently available and the type of predominantly adopted (ecological) study design, most epidemiological studies available are not able to give an exhaustive answer to the question whether and to what extent air pollution increases the COVID-19 disease severity.

In Italy, the National Health Institute of Health (ISS) and the National Network System for Environmental Protection (SNPA-ISPRA) have launched, in collaboration with the Italian Environment and Health Network (https://rias.epiprev.it/) (accessed on 24 February 2022), the EpiCovAir epidemiological studies program, based on the data produced by the national integrated COVID-19 surveillance (https://www.epicentro.iss.it/coronavirus/) (accessed on 20 February 2022) and by the SNPA (www.snpambiente.it) (accessed on 20 February 2022). EpiCovAir aims to carry out epidemiological studies at the national level to verify the association of long-term exposure to air pollution and the onset of symptoms and the severity of the health effects among COVID-19 cases in Italy while adjusting for socio-demographic and economic confounding factors associated with the infection.

In this regard, there is a need to better understand the role of contextual variables such as municipalities characteristics, resident population features, the network of social relations, the mobility of people, and the responsiveness of the NHS. It is also necessary to synthesize the content of spatial information, available at the municipal level, in a smaller set of “summary indexes” variables that can be more easily viewed and analyzed for understanding the dynamics of the phenomenon.

The objective of this paper is therefore to identify and collect the contextual factors available at the municipality level in Italy (characteristics of administrative areas, demographic, mobility, and socio-economic-health information of communities), and to synthesize their information content to produce indicators to be used in national epidemiological studies aimed at assessing the role of air pollution on the incidence and severity of the COVID-19 disease.

## 2. Materials and Methods

### 2.1. Variables Selection

A large number of spatial variables related to the 7903 Italian municipalities (list as of 1 January 2020—ISTAT) and their resident communities, were initially identified, collected and processed, and grouped into five information dimensions a priori defined as: (1) geographic characteristics of the municipality, (2) demographic and anthropogenic characteristics, (3) mobility, (4) socio-economic-health area, and (5) availability of health care.

Most of the data used in the analyses carried out in this paper are freely downloadable, and Appendix A shows the complete list of the variables, their description, the temporal dimension, and the source of the data. The spatial typology of data collected varies from municipality to municipality and are assumed to be constant over the period of time to which they refer.

Due to the high correlation of several variables referring to the same phenomenon (for example, altitude and altimetric zone), in order to avoid redundancy of information, 44 variables by five macro-categories were selected; three further variables were added to describe pollution levels for each Italian municipality.

### 2.2. Statistical Analysis

In order to evaluate the relationship between the variables within each dimension, and to identify the presence of further redundancy, Spearman correlation coefficients (ρ) have been estimated to quantify the relationship between each variable pair (x and y). Conventionally, values of ρ between 0 and 0.19 indicate absence of correlation, values of ρ between 0.2 and 0.39 indicate weak correlation, values of ρ between 0.40 and 0.59 are indices of a moderate correlation, values of ρ between 0.6–0.79 represent a high correlation, and finally values of ρ higher than 0.8–1.0 are indicators of a very high correlation. 

A principal component analysis (PCA) was performed for each dimension. The goal of the PCA is to reduce the large number of initial variables into a smaller number of components, limiting as much as possible the loss of information content (variability). This occurs through a linear transformation of the variables that projects the original ones into a new Cartesian system in which the variables are sorted in decreasing order of variance. Therefore, the variable with the greatest variance is projected to the first axis, the second to the second axis, and so on. The reduction of complexity occurs by limiting itself to analyzing the main ones (in terms of variance) among the new variable. The PCA is effective only when there is a good share of variance in common among the variables (with correlation coefficients that are not very low or very high); in this case, a few principal components will be sufficient to obtain a good approximation to the starting matrix. The advantage of the PCA is the ability to condense most of the variances and covariances present in the initial set of variables into the first components. Thus, considering only the first principal components, we obtain the best possible synthesis of the information provided by the initial variables.

Within each of the five dimensions a priori defined, each main component represents a linear combination of the starting variables and, consequently, the intra-group correlation between the components is equal to 0.

For the PCA purposes, the starting variables were therefore standardized (mean = 0 and variance = 1); then, for each dimension, only the main components with eigenvalues ≥ 1 were selected [22]. This guideline is based on the idea that, given a certain total variability of all standardized variables, a PCA should explain at least one variation equal to the mean value of a single standardized variable. 

The analyses also included ordinal qualitative variables for which it made sense to hypothesize a unit linear increase in the transition from one category to another (for example, degree of urbanization, socio-economic position (SEP).

We also analyzed correlation of PCA components with PM_2.5_, PM_10_ and NO_2_ population weighted exposure (PWE) values, obtained using a four-stage approach based on the machine learning method, “random forest”, which use space–time predictors, satellite data, and air quality monitoring data estimated at a national level [23].

All analyses were conducted using R statistical software (version 3.6.0) [24].

## 3. Results

Table 1 describes the characteristics, across the 7903 Italian municipalities, of the 18 contextual continuous variables for each dimension under study and air pollution PWE concentrations, while Table 2 shows the distribution of the 7903 Italian municipalities with respect to the 26 contextual categorical variables for each dimension under study.

The area of the Italian municipalities varies from 0.120 km^2^ to about 1300 km^2^ (mean 35.2; standard deviation (SD) 50.8). The maximum altitude is 2035 m above sea level. Coastal municipalities account for 8.1% of the total, while 0.4% are island municipalities. About 64% of the municipalities are located in rural or sparsely populated areas. As regards the level of anthropization of the Italian municipalities, we used the maximum value of impervious surfaces (ISA) in a 1 km × 1 km cell within the municipal area. ISA is an indicator of the spatial distribution of surfaces. Examples of ISAs include streets, parking lots, buildings, driveways, sidewalks. The ISA maximum value measured in a 1 km × 1 km cell was equal to 58.6 (SD = 42.7), while for the night luminosity index, the measured value was 23.1 (SD = 41.9). The percentage of urban coverage (maximum value in a 1 km × 1 km cell) is on average less than 50% (41.2% SD = 27.6).

The population as of 31 December 2019 ranged from a minimum of 30 to a maximum of about 3 million inhabitants, with a median of 2459 residents. The median population density is 105 inhabitants per km^2^. The percentage of residents aged 65 years and more in the Italian municipalities ranges from a minimum of 8.6% to a maximum of 62.3% (95th percentile equal to 34.8%).

As for mobility, the median value of the attraction index is equal to 21.1 (5th percentile 6.7, 95th percentile 45.7), while for the self-containment index we observed a value equal to 33.2 (5th percentile 15.5, 95th percentile 59.7). The maximum number of people who move outside the municipality for work or study reasons is more than 90,000 people, while about 1,300,000 move within the municipality. In 41.5% of the municipalities, there is an airport within 30 km of the municipal boundaries (4.8% 2 or more), while only 22.3% of the municipalities have at least one railway station on its territory.

For the variables describing the socio-economic-health dimension, the average family income in Italian municipalities is equal to 13,000 Euros (SD 3123; min 3796; max 29,985), while the entrepreneurship rate varies from a minimum value of 9 to a maximum of 407 companies per 100,000 inhabitants. Annual all causes mortality rates vary between 0.7 (5th percentile) and 2 (95th percentile) percentile) with a mean 1.2 (SD 0.4) per 100; cardiovascular disease mortality rates vary between 0.2 (5th percentile) and 0.9 (95th percentile) with a mean of 0.5 (SD 0.2) per 100; mortality rates from respiratory diseases vary between 0.02 (5th percentile) and 0.2 (95th percentile) with a mean of 0.1 (SD 0.06) per 100. Annual hospitalization rates of residents in Italian municipalities vary between 4 (5th percentile) percentile) and 6.1 (95th percentile) with a mean of 5 (SD 0.2) per 100; cardiovascular disease hospitalization rates vary between 0.8 (5th percentile) and 1.8 (95th percentile) per 100 residents with an average of 1.2 (SD 0.31) per 100 residents; and respiratory disease hospitalization rates vary between 0.5 (5th percentile) and 2 (95th percentile) with an average of 0.7 (SD 0.2) per 100.

Regarding health care availability, the minimum average distance of Italian municipalities from a health facility is just over 9.4 km (SD 6 km), while the distance from an emergency room is 10.7 km (SD 6.4 km). Less than 1% of municipalities (0.91%) have at least one hospital or university hospital on their territory, approximately 7% have at least one hospital, and 3.6% of municipalities have at least one nursing home. In 8.2% of the municipalities, at least one acute care bed is available. In 4% of the municipalities, at least one bed is available for long-term care; in 5.3%, at least one bed is available for rehabilitation; and in 4.8%, at least one place is available in intensive care (3.2% between 1 and 10 places). In 230 municipalities (2.91% of the total), there is at least one emergency room, and in 589 municipalities (7.5%), at least one private hospital. In the territory of 481 municipalities (6%), there is at least one residence for the elderly with medical assistance.

As for air pollution, in the period 2016–2019, the population weighted average exposure value to PM_2.5_ was equal to 14.6 µg/m^3^ (SD 5.0 µg/m^3^), for PM_10_ it was 21.1 µg/m^3^ (SD 6.5 µg/m^3^) while for NO_2_ it was equal to 14.5 µg/m^3^ (SD 6.5 µg/m^3^) with a maximum of 46.3 µg/m^3^.

Figure 1 shows the correlations between the variables within each of the five categories under study.

Among the geographic characteristics of the municipalities (Figure 1a), the greatest correlation is observed between the coastal zone and coastal municipality variables (ρ = 0.72). For the demographic and anthropogenic characteristics dimensions (Figure 1b), the percentage of population over-65 years shows a moderate negative correlation with all the other variables, which in general are all very correlated to each other. For the mobility dimension (Figure 1c), a correlation equal to 0.97 is observed between total movements and intra-municipal and extra-municipal movements, which are highly correlated (also between them) (*p* = 0.88). From the correlation matrix of the socio-economic-health dimension (Figure 1d), the all-cause mortality rate is highly correlated with that for diseases of the circulatory system (ρ = 0.66), and the total hospitalization rate with that for diseases of the circulatory system (ρ = 0.75).

As for the health supply dimension (Figure 1e), the variables that describe the supply of beds are all positively correlated with each other and also positively correlated with the presence of health facilities in the municipal area. Finally, the variables distance from an emergency room and distance from a health facility are highly correlated (ρ = 0.85).

The results of the PCA are shown in Figure 2. For each dimension, the most informative components are displayed (i.e., components with eigenvalue ≥ 1). Size and colors of dots indicate the contribution of the single variable in the explanation of the component; in this way, in fact, it is possible to understand which variable contributes most to the construction of the component itself. Overall, the PCA made it possible to extract twelve components: three for the territorial characteristics dimension of the municipality (variance explained 72%), two for the demographic and anthropogenic characteristics dimension (variance explained 62%), three for the mobility dimension (variance explained 83%), two for the socio-economic-health dimension (variance explained 58%), and two for the health supply dimension (variance explained 72%).

All the components of the different dimensions are only marginally correlated with each other (Figure 3), demonstrating their potential ability to grasp different aspects of the geographical distribution of the COVID-19 pathology.

Table 3 shows the correlation of the twelve components with the air pollution variables. We did not find high correlation values between components and the Italian population weighted exposure level to PM_2.5_, PM_10_, and NO_2_, except for the second component of the territorial characteristics dimension of the municipality for which there is a high negative correlation with all three pollutants.

## 4. Discussion

A large body of evidence on the impact of air pollution on human health has accumulated over the past years. Among the various documented health effects, air pollution also increases the risk of chronic diseases (respiratory, cardio-metabolic), described as the comorbidities increasing the risk of being hospitalised or dying from COVID-19. This evidence contributed to a fast proliferation of epidemiological studies linking ambient air pollution to COVID-19 disease. However, the exact role of air pollutants in modulating the spread and severity of COVID-19 is still unclear.

Among the aspects contributing to make the issue difficult to face, there is the need to integrate methods and approaches belonging to different disciplines, like epidemiology of infectious diseases, environmental epidemiology of non-communicable diseases, and assessment and modelling of exposure to air pollutants. To correctly address the question of how and how much air pollution does impact on COVID-19 disease, it entails both to understand the spatial and temporal dynamics of the epidemics, whose spread is primarily based on direct contagion, and to identify the most relevant factors linked to the probability of becoming a case, and/or to the risk of hospitalisation and disease prognosis.

Dealing with the above aspects strongly depends, among other aspects, on the ability to adopt appropriate study design/analytical models, and to have information on individual variables (age, gender, comorbidities, etc.) and on contextual covariates. Contextual factors include characteristics of the area of residence, socio-economic indicators, availability of healthcare services and social interactions within communities, mobility, time-activity patterns, type of environment (urban, rural, semi-rural), prevalent occupational activities, and demographic and background health profiles.

Epidemiological research has a long tradition of studies based on the systematic reports about potential time and space varying determinants of diseases (i.e., characteristics of the territory and of the population), following the pioneering work of William Farr with his studies on cholera [25].

The present work follows this approach by addressing the critical issue of how to identify, collect and synthetize relevant information from large national datasets of contextual variables, in order to better characterise, through epidemiological studies, the relationships between air pollution and COVID-19 contagion and severity at the national level.

By applying data reduction techniques to the overall dataset of collected variables, this work made it possible to identify few informative summary factors accounting for large amounts of the observed variance within each of the five dimensions of contextual factors: from 58% (socio-economic-health dimension) to 83% (mobility dimension). The explained variance might be greatly increased by including one or two additional components with eigenvalues approaching the value of 1. For instance, regarding the socio-economic-health dimension (58% explained variance), the PCA analyses showed that including a third component with eigenvalue = 0.96, and a fourth component with eigenvalue = 0.95, would increase the explained variance respectively to 68% and to 79%. Similarly, the inclusion of a third component with eigenvalue = 0.91 would augment the healthcare offer dimension’s overall explained variance from 72% to 79%.

The ongoing EpiCovAir program, as well as other epidemiology studies, would benefit from the explorative and preparatory work herein presented. The methods adopted are of course still susceptible to improvement. To this regard, new analyses are focusing, for instance, on the use of a generalized propensity score (GPS) approach [26], representing the conditional probability of being exposed to air pollution given the observed values of area-level covariates, to account for the major determinants of the spatial distribution of COVID-19 cases and case-fatality rates.

## 5. Conclusions

In conclusion, the present work provides both a method and a dataset to be used in epidemiological studies on the association between chronic exposure to air pollution and health outcomes in Italian territory. All information has been collected and is available at the municipality level. The data repository is available upon request to the authors.

## Figures and Tables

**Figure 1 ijerph-19-02859-f001:**
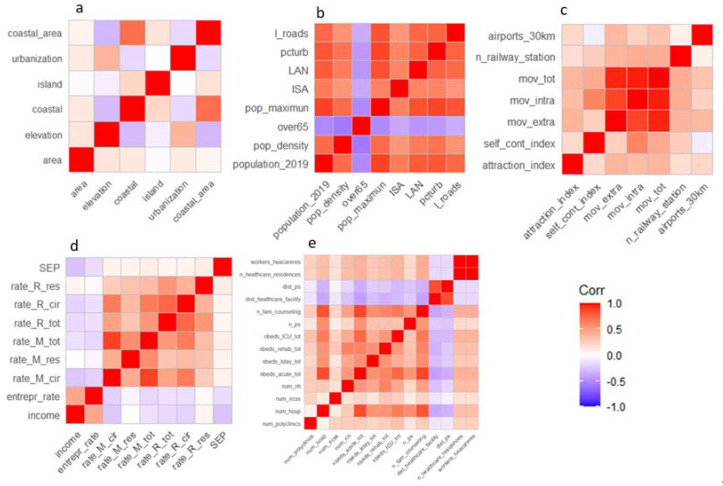
Spearman correlation matrices among the variables within each dimension: geographic characteristic (**a**), demographic and anthropogenic characteristics (**b**), mobility (**c**), socio-economic-health characteristics (**d**), availability of health care (**e**). Variable labels are listed in Table 1.

**Figure 2 ijerph-19-02859-f002:**
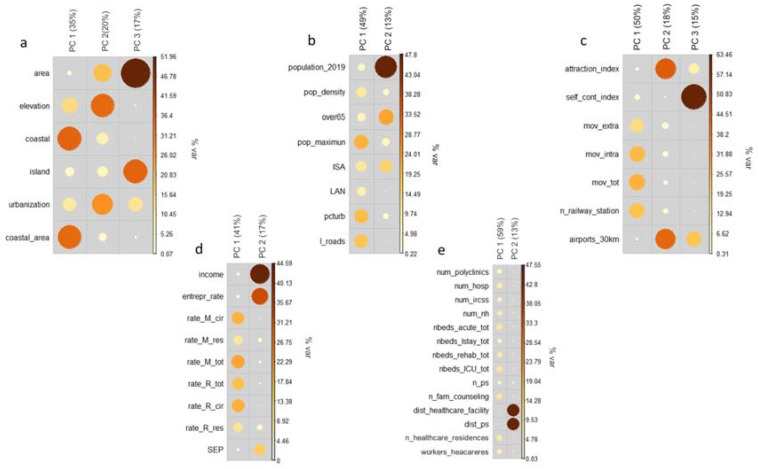
Results of the principal component analysis (PCA); the contributions of individual contextual covariates on the selected components are displayed for the five dimensions (geographic characteristic (**a**), demographic and anthropogenic characteristics (**b**), mobility (**c**), socio-economic and health status of the population (**d**), availability of health care (**e**)). Variable labels are listed in Table 1.

**Figure 3 ijerph-19-02859-f003:**
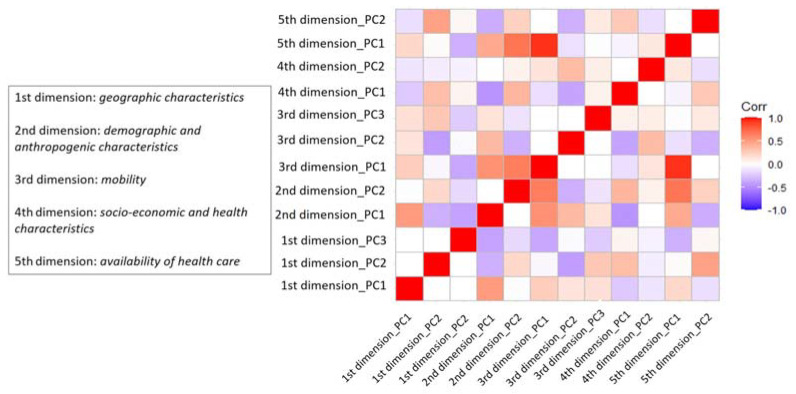
Correlation matrix among the components resulted from the principal component analysis.

**Table 1 ijerph-19-02859-t001:** Characteristics # of the contextual continuous variables for each dimension under study and air pollution exposure across the 7903 Italian municipalities.

Variables	Label	Mean	SD	Min	p5	p25	p50	p75	p95	Max
**1st dimension: geographic characteristics**										
Area (km^2^) as of 1 January 2020	area	38.2	50.8	0.12	4.37	11.5	22.4	44.6	125.7	1287.4
Altitude (m above sea level)	elevation	355	296	0	12	114	289	520	920	2035
**2nd dimension: demographic and anthropogenic characteristics**										
Resident population as of 31 December 2019 (number)	population_2019	7622	42,801	30	284	1005	2459	6317	25,090	2,837,332
Population density ratio (population over area)	pop_density	303.7	649.7	0.80	12.6	43.5	105.5	281.5	1189	12,178
Population density (cell 1 km^2^)	pop_maximun	1888	2444	12	165	509	1079	2,316	6429	35,271
Percentage of population over-65 years as of 31 December 2019 (%)	over 65	25.5	5.40	8.64	17.7	21.9	24.9	28.3	34.8	62.3
Impervious Surface Areas (cell 1 km^2^)	ISA	58.6	42.7	0	0	0	79	100	100	101
Value of the night brightness index (cell 1 km^2^)	LAN	23.1	41.9	0	2.22	7.28	15.2	28.5	63.6	1013
Percentage of urban coverage (cell 1 km^2^)	pcturb	41.2	27.6	0	0	22.2	37.6	59.7	94.4	100
Length of the roads (cell 1 km^2^)	l_roads	12,332	5551	1673	5426	8382	11,337	15,134	22,477	51,711
**3rd dimension: mobility**										
Attraction index (mean 2014–2015) *	attraction_index	23.2	12.2	0	6.69	13.8	21.4	30.7	45.7	83.1
Self-containment index (mean 2014–2015) **	self_cont_index	34.7	13.3	2.48	15.5	25.3	33.2	42.5	59.7	89.1
Extra-municipal movements ***	mov_extra	1439	2633	1	79	267	638	1654	5,127	90,063
Intra-municipal movements ****	mov_intra	2214	18,293	0	24	163	483	1382	6793	1,284,994
Total movements: number of individuals who move for work or study	mov_tot	3653	20,008	5	118	457	1181	3126	11,836	1,340,818
**4th dimension: socio-economic and health characteristics**										
Household income (average 2014–2015 in €) *****	income	13,000	3124	3796	8037	10,285	13,453	15,245	17,612	29,985
Entrepreneurship rate (2014–2015 average): number of companies per 100,000 res.	entrepr_rate	62.6	24.2	9.64	33.3	47.8	59.5	73.1	99.6	407.4
Cardiovascular diseases hospitalization rate (annual average 2013–2018 per 100 residents)	rate_R_cir	1.18	0.31	0.17	0.80	0.97	1.12	1.31	1.76	4.37
Respiratory diseases hospitalization rate (annual average 2013–2018 per 100 residents)	rate_R_res	0.69	0.18	0	0.45	0.58	0.67	0.78	1.00	2.82
All causes hospitalization rate (annual average 2013–2018 per 100 residents)	rate-R_tot	4.95	0.67	1.06	4.03	4.51	4.88	5.29	6.14	12.3
Cardiovascular diseases mortality rate (annual average 2013–2017 per 100 residents)	rate_M_cir	0.46	0.22	0	0.20	0.31	0.42	0.55	0.85	2.37
Respiratory diseases mortality rate (annual average 2013–2017 per 100 residents)	rate_M_res	0.09	0.06	0	0.02	0.05	0.08	0.11	0.20	1.06
All causes mortality rate (annual average 2013–2017 per 100 residents)	rate_M_tot	1.19	0.43	0	0.66	0.89	1.12	1.38	1.97	5.62
**5th dimension: availability of health care**										
Minimum distance between the municipality (centroid) and a health facility (meters)	dist_healthcare_facility	9403	5987	65	1753	5,228	8428	12,635	20,122	152,024
Minimum distance between the municipality (centroid) and an emergency room (meters)	dist_er	10,751	6382	57	2470	6274	9733	14,050	22,206	151,546
Number workers in healthcare residences	workers_heacareres	217	566	2	13	34	71	172	736	5940
**Air Pollution levels**										
Population weighted exposure of PM_2.5_ (annual mean 2016–2019 µg/m^3^)	pm25_2016_2019_pop	14.6	4.98	6.11	8.48	10.5	12.9	19.1	23.4	27.4
Population weighted exposure of PM_10_ (annual mean 2016–2019 µg/m^3^)	pm10_2016_2019_pop	21.1	6.46	6.62	11.8	16.0	20.1	26.1	32.6	37.5
Population weighted exposure of NO_2_ (annual mean 2016–2019 µg m^3^)	no2_2016_2019_pop	14.5	6.74	4.23	6.38	8.73	13.0	19.3	26.1	46.3

# mean, standard deviation (SD), percentiles (p5, p25, p50, p75, p95), minimum (min) and maximum (max) value; * number of non-resident individuals who carry out work or study activities in the municipality over the total mobility flows (active residents plus outgoing flows of residents); ** number of resident individual who carry out work or study activities in the municipality over the total mobility flows (active residents plus outgoing flows of residents); *** number of individuals who travel outside the municipality of residence for work or study reasons; **** number of individuals who move within the municipality of residence for work or study; ***** ratio between the total gross income of registered households and the total number of members of registered households.

**Table 2 ijerph-19-02859-t002:** Distribution of the 7903 Italian municipalities with respect to contextual categorical variables for each dimension under study.

Variable	Label	Number of Municipalities	%
**1st dimension: geographic characteristics**			
Coastal municipality	coastal	642	8.12
Island municipality	island	34	0.43
Coastal area *	coastal_area	1165	14.7
Degree of urbanization:			
Cities or “densely populated areas”	urbanizzaztio_1	255	3.23
Small towns and suburbs or “intermediate population density areas”		2607	33.0
Rural areas or “sparsely populated areas”		5041	63.8
**3rd dimension: mobility**			
Number of airports within 30 km from the municipality boundaries	airports_30 km		
1		2891	36.6
2 or more		386	4.88
Number of railway stations in the municipality	n_railway_station		
1		1296	16.4
2–3		387	4.89
4–5		57	0.72
6 or more		29	0.33
**5th dimension: socio-economic and health characteristics**			
Socio economic position (SEP) **	SEP_cat		
Low		1550	19.6
Middle-low		1582	20.0
Middle		1610	20.4
Middle-high		1592	20.1
High		1569	19.9
**6th dimension: availability of health care *****			
Number of teaching hospitals	num_polyclinics		
1		27	0.34
2–3		12	0.45
3 or more		6	0.12
Number of general hospitals	num_hosp		
1		518	6.55
2–3		22	0.28
3 or more		7	0.09
Number of public or private foundations	num_ircss		
1		38	0.48
2–3		5	0.07
3 or more		3	0.03
Number of accredited private nursing homes	num_nh		
1		220	2.78
2–3		48	0.61
3 or more		17	0.19
Number of acute care beds	nbeds_acute_ord		
1–10		19	0.24
10–50		127	1.61
51–150		235	2.97
150 or more		269	3.40
Number of long-term hospital beds	nbeds_lstay_ord		
1–10		99	1.25
10–50		183	2.32
51–150		31	0.39
150 or more		5	0.06
Number of rehabilitation beds	nbeds_rehab_ord		
1–10		61	0.77
10–50		214	2.71
51–150		114	1.44
150 or more		30	0.38
Number of intensive care beds	nbeds_ICU_ord		
1–10		254	3.21
10–50		83	1.05
51–150		14	0.18
150 or more		3	0.04
Number of emergency department	n_ps		
1		219	2.77
2–3		7	0.09
3 or more		4	0.05
Number of family counseling	n_fam_counseling		
1		436	5.52
2–3		123	1.56
3 or more		30	0.38
Number of nursing residences	n_healthcare_residences		
At least 1		481	6.09

* municipalities with at least 0% of the surface at a maximum distance of 10 km from the sea; ** data from the 2011 Italian Census and calibrated on a regional basis; *** on 31 December 2019.

**Table 3 ijerph-19-02859-t003:** Spearman correlation coefficients among pollution variables and components resulted from the principal component analysis for the five dimensions (geographic characteristic, demographic and anthropogenic characteristics, mobility, socio-economic and health status of the population, availability of health care).

PCA Dimension	PM_2.5_	PM_10_	NO_2_
**Geographic characteristic**			
First component	0.48	0.62	0.50
Second component	−0.80	−0.79	−0.77
Third component	0.05	0.01	−0.11
**Demographic and anthropogenic characteristics**			
First component	0.42	0.51	0.58
Second component	−0.35	−0.36	−0.33
**Mobility**			
First component	0.37	0.43	0.51
Second component	0.47	0.46	0.58
Third component	−0.20	−0.15	−0.20
**Socio-economic and health status of the population**			
First component	−0.44	−0.46	−0.53
Second component	0.29	0.17	0.36
**Availability of health care**			
First component	0.45	0.45	0.57
Second component	−0.47	−0.45	−0.57

## Data Availability

The data presented in this study are available on request from the corresponding author.

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
