# Peer review of "A Methodological Approach to Use Contextual Factors for Epidemiological Studies on Chronic Exposure to Air Pollution and COVID-19 in Italy"

_ijerph, 2022, doi:10.3390/ijerph19052859_

Round 1

Reviewer 1 Report

First of all, I would like to comment that the work carried out by the authors seems to me to be enormous due to the scope of the study.

Gathering all the data and putting it on paper and not getting lost in the data is a huge undertaking.

It is true that the Covid-19 data is very important, however with these data many more conclusions can be drawn.

The authors refer that there are few works, specifically the authors say: This work faces for the first time, to our knowledge, the critical problem of how to identify, collect and synthesize relevant information from large national data sets..."

I invite you to review the works of William Farr and his work around the 1,840 epidemiological studies of cholera in London. Perhaps I am missing this reference that he is undoubtedly the pioneer in this type of work with the media of that time and the conclusions that he achieved without the means that we have today that allow us to make such beautiful graphics.

Obviously, the work presented by the authors is not usually done due to the complexity involved and the amount of metadata that is required.

Even so, the work is well structured, it is clear and shows clear results.

I hope that this work not only serves for this work but that they manage to extract many more epidemiological results from their data. Data on asthma, allergies, and other pollutants may be relevant and visible in future articles.

Author Response

We would like to thank the reviewer for his positive review and suggestions.

We have added among the references the work of William Farr “Report on the Mortality from Cholera in England, 1848–1849. London: HMSO, 1852” and we added in the text the sentence “Epidemiological research has a long tradition of studies based on the systematic reports about potential time and space varying determinants of diseases (i.e. characteristics of the territory and of the population), following the pioneering work of William Farr with his studies on cholera”.

Reviewer 2 Report

In manuscript ID: ijerph-1597956 , the authors identify and collected the municipalities and community contextual factors and to synthesize their information content to produce suitable indicators in national environmental epidemiological studies, with specific emphasis on assessing the possible role of air pollution on the incidence and severity of the COVID-19 disease.

The work is of interest, and scientifically sound. The manuscript does not present particular errors or omissions. I have any major concerns. Please check the references style and format

Author Response

We thank the reviewer for his positive feedback.

We have checked the references style and format, as appropriately suggested

Reviewer 3 Report

Please number the lines, minor spell check, match the references with the text.

In general interesting and original idea, the approach and the statistical analysis are well performed.

In your future analysis , take into account the working environment of the population , because it could be  also a risk factor for the diseases you  already described , being at the end a confounding factor.

Author Response

Thanks for the useful suggestions.

We have aligned text and references.

We have not considered working environment in this first dataset at municipal level, but we thank the reviewer for the useful suggestion to be taken into consideration for further development of our work.